# Eco-Friendly Reduction of Graphene Oxide by Aqueous Extracts for Photocatalysis Applications

**DOI:** 10.3390/nano12213882

**Published:** 2022-11-03

**Authors:** Luz H. Verástegui-Domínguez, Nora Elizondo-Villarreal, Dora Irma Martínez-Delgado, Miguel Ángel Gracia-Pinilla

**Affiliations:** 1Materiales Nanoestructurados (CICFIM), Facultad de Ciencias Físico Matemáticas (FCFM), Universidad Autónoma de Nuevo León (UANL), 66450 San Nicolás de los Garza, N.L., Mexico; 2Mesoscale Chemical Systems, MESA+ Institute, University of Twente, P.O. Box 217, 7500AE Enschede, The Netherlands

**Keywords:** eco-friendly reduction, reduced graphene oxide, aqueous extracts, photocatalysis

## Abstract

In the present work, reduced graphene oxide was obtained by green synthesis, using extracts of *Larrea tridentata* (gobernadora) and *Capsicum Chinense* (habanero). Graphene oxide was synthesized by the modified Hummers’ method and subsequently reduced using natural extracts to obtain a stable and environmentally friendly graphene precursor. Consequently, the gobernadora aqueous extract was found to have a better reducing power than the habanero aqueous extract. This opportunity for green synthesis allows the application of RGO in photocatalysis for the degradation of the methylene blue dye. Degradation efficiencies of 60% and 90% were obtained with these materials.

## 1. Introduction

Nowadays, as a consequence of industrialization, the world faces problems with air, soil, and water pollution. This problem is becoming increasingly important despite the existence of pollutant removal treatments and waste reduction and disposal programs. In many of these processes, the problem is only partially eliminated, so pollution of rivers and seas is increasing, and the damage this is causing is of an unimaginable magnitude. In the last two decades, the organic pollutants discarded by industries have no control, which has led to water pollution being severely affected. The World Bank estimates that 17–20% of industrial water pollution comes from textile dyeing. Wastewater from textile industries is a major problem for conventional treatment plants worldwide. The release of these wastewaters into natural environments is very problematic for aquatic organisms. The wastewater generated by the textile industries is known to contain considerable quantities of non-fixed dyes, especially azo dyes, and a large number of inorganic salts. It has been estimated that more than 10% of the total dye is used in dyeing processes while the rest is released to the environment. Therefore, it is necessary to study new processes that allow the efficient elimination of pollutants, which are affecting the environment so much [1,2,3,4].

Several methods are used to treat different dye contamination, such as membrane processes, ultrafiltration, adsorption, biodegradation by micro-organisms, advanced oxidation and photocatalytic degradation [3,4,5]. Among these methods, the photocatalytic approach gained great attention among scientists and the industrial sector, due to the use of solar energy for the removal of dyes. This is due to its promising, effective and efficient pollutant degradation activity, which is produced by allowing spontaneous and non-spontaneous reactions to optimize the entire process; it can be implemented with low-cost catalysis [6]. Many nanomaterials and nanocomposites are being used as catalysts for the removal of various dyes, such as nanoparticles of ZnO [7], TiO_2_ [8], copper [9], nanocomposites TiO_2_Co_3_O_4_ [10], graphene [11,12,13] and others.

Graphene, an extremely attractive component, has been particularly chosen in photocatalysis due to its large specific surface area and an extensive two-dimensional conjugation π and π* structure that has excellent electronic conductivity [7,8,9,10,11,12,13,14], is used for the photodegradation of different dyes which include methylene blue, methyl orange and rhodamine B [7,8,9,10,11,12,13,14]. However, one of the technical difficulties of this novel material is mass production. Currently, conventional chemical synthesis methods use toxic compounds such as hydrazine or sodium borohydride [15] for the production of reduced graphene oxide (RGO), which causes the accumulation of wastes harmful to the environment and human health. Therefore, one of the big challenges is the production of RGO from products with greater environmental compatibility, such as natural agents, plant extracts, glucose, etc. Few plants are being used as natural reducing agents for the reduction of graphene oxide [16,17,18,19,20], therefore, it is important to provide new data that allow the use of natural products to obtain this material.

The present research focuses on the synthesis of reduced graphene oxide using natural extracts of *Larrea Tridentata* (gobernadora) and *Capsicum Chinense* (habanero). The possible route of chemical reactions in the modified Hummers’ method [21] of graphite oxide and the physicochemical properties of synthesized RGO are discussed and confirmed by different characterization techniques. In addition, RGO is used as a photocatalyst material to remove methylene blue dye in an aqueous solution. The degradation of MB dye developed through ultraviolet radiation and degradation mechanisms were discussed.

### Photocatalysis

The photocatalysis has emerged as a strong candidate for practical field applications. The process relies on the oxidation capacity of the photocatalytic process to break down resilient pollutants into their corresponding simpler forms. The utilization of the photocatalytic degradation of the methylene blue (MB) as the model reaction for estimating the photocatalytic capabilities of the novel engineered nanomaterials has become a common practice in recent times. Some of the considerations that have contributed towards the excessive usage of this reaction include (a) MB is a common carcinogenic pollutant and its removal from the aqueous medium is highly desirable, (b) MB dye is ultraviolet visible (UV–Vis) active species and exhibits characteristic peaks in the UV–Vis spectrum, (c) it is a colored dye and exhibits the blue color in its oxidized state. Therefore, the degradation of MB in any medium can also be visually observed [22]. The general mechanism of photocatalytic degradation is illustrated in Figure 1. Photocatalysis starts with the transport of contaminants from the surroundings to the photocatalyst surface. The dye compounds are first adsorbed onto the surface on which oxidation–reduction reactions occur. These redox reactions are forced by photogenerated electrons in the conduction band (CB) and holes in the valence band (VB). The fragments of dyes are degraded further into reaction intermediates, including aldehydes, carboxylic species, phenols and amines which are ultimately converted into H_2_O, CO_2_, ammonium ions and sulfate ions [23].

## 2. Materials and Methods

The materials used were potassium permanganate, reactive ACS, (≥99.0 Sigma-Aldrich (St. Louis, MO, USA) (KMnO_4_), deionized H_2_O, sulfuric acid (H_2_SO_4_) 95–98%, 3% hydrogen peroxide and graphite powder from Sigma Aldrich.

### 2.1. Synthesis of Graphene Oxide

The graphite was oxidized by the modified Hummers’ method [21]. A mixture of 3 g of graphite was prepared with 70 mL of a 0.5 M H_2_SO_4_ solution under constant stirring in a cold bath until a temperature of 0 °C was reached. Slowly 9 g of KMnO_4_ was added, and minutes later the mixture was removed from the cold bath to immediately incorporate an oil bath system at 35 °C under constant agitation. After 1 h 125 mL of deionized water was slowly added to the system, causing the temperature to rise, keeping the system at that temperature for a period of 15 min. Subsequently, 15 mL of 3% hydrogen peroxide was added to the mixture and heated to 60 °C, to reduce the KMnO_4_ residue, now becoming a brownish-yellow mixture. Finally, the mixture was filtered and washed with deionized water repeatedly until reaching a neutral pH. Once the oxide of the reaction was obtained, the suspension was prepared at a concentration of 12 mg/mL and then it was introduced in an ultrasound bath for 10 h, to obtain GO exfoliation.

#### Modified Hummers’ Method

Below is a possible route of chemical reactions in the modified Hummers’ method:

Sulfuric acid is strong enough to produce bisulfate, which reacts with sp^2^ bonds present in graphite [24].
(1)$$2H_{2}{\mathrm{SO}}_{4}\to {\mathrm{HSO}}_{3}^{+}+{\mathrm{HSO}}_{4}^{-}$$
(2)$$H_{2}O+H_{2}{\mathrm{SO}}_{4}\;↔{\;H}_{3}+{\mathrm{HSO}}_{4}^{-}$$

Bisulfate intercalates graphite with sulfuric acid. This intercalation is complete in the presence of an oxidizing agent [25]:(3)$$\mathrm{GIC}’s:{\;C}_{24s}^{+}\cdot {\mathrm{HSO}}_{4}^{-}2H_{2}{\mathrm{SO}}_{4}$$
where “GIC” refers to the intercalation of the graphite and “s” is the number of stages.

In the Hummers Method, the functional state of sodium nitrate as an oxidizing agent to form graphite bisulfate was modified by the following reactions:(4)$$\;5C_{24s}+{\mathrm{KMnO}}_{4}+17H_{2}{\mathrm{SO}}_{4\;}\to {\;C}_{24}^{s}{\mathrm{HSO}}_{4}^{-}2H_{2}{\mathrm{SO}}_{4}+{\mathrm{MnSO}}_{4}+{\mathrm{KHSO}}_{4}+4H_{2}O$$

The experimental data reported on the above equation for stages (1)–(4) of graphite bisulfate. They can be lost in the form of gases in the following way [25]:(5)$$5C+4{\mathrm{KMnO}}_{4}+8H_{2}{\mathrm{SO}}_{4}\;\to 5{\mathrm{CO}}_{2}+4{\mathrm{MnSO}}_{4}+4{\mathrm{KHSO}}_{4}$$

Another reaction that takes place with the mixture of sulfuric acid and permanganate at 0 °C that does not appear in the formation of the principal gases is in the following equation:(6)$$2{\mathrm{KMnO}}_{4}+2H_{2}{\mathrm{SO}}_{4}\to {\mathrm{Mn}}_{2}O_{7}+2{\mathrm{KHSO}}_{4}+H_{2}O$$

In this stage of the reaction, we have the following compounds GIC’s, ${\mathrm{MnSO}}_{4}$, $H_{2}O$, ${\mathrm{Mn}}_{2}O_{7}$, in addition to concentrate sulfuric acid and permanganate that do not make room for the formation of GIC. To continue with the oxidation of graphite once the GIC was obtained the solution’s temperature changes from 0 °C to 35 °C. The permanganate behavior in the presence of sulfuric acid in a range of 20–35 °C is described in the following reactions [26,27,28]. In the Figure 2, shown of the permanganate behavior in the presence of sulfuric acid:(7)$${\mathrm{KMnO}}_{4}+{\;H}_{2}{\mathrm{SO}}_{4}\to {\;K}^{+}+{\mathrm{MnO}}_{3}^{+}+H_{3}O^{+}+3{\mathrm{HSO}}_{4}^{-}$$

O well
(8)$$2{\mathrm{KMnO}}_{4}+2H_{2}{\mathrm{SO}}_{4}\to {\mathrm{Mn}}_{2}O_{7}+H_{2}O+2{\mathrm{KHSO}}_{4}$$

${\mathrm{MnO}}_{4}^{-}\;$in the presence of air can generate in a reversible equation, returning to ${\mathrm{Mn}}_{2}O_{7}$ in the presence of concentrated sulfuric acid as dictated by the following equation [27]:(9)$${\mathrm{Mn}}_{2}O_{7}+H_{2}O\;↔2H^{+}+2{\mathrm{MnO}}_{4}^{-}$$

By adding deionized water we believe that in the reaction more ${\mathrm{MnO}}_{4}^{-}$ ions are produced on ${\mathrm{Mn}}_{2}O_{7}\;$at the same time that an exothermic reaction is generated, due to the ions that we have in the solution, activation of permanganate ions as a result, the reaction happens from a dark color to a brown color as a sign that the graphite is oxidized.

The solution is maintained at 98 °C for 15 min to continue its oxidation. When added $H_{2}O_{2}$ reacts with the permanganate ions in the following way:(10)$${\;H}_{2}O_{2}+{\mathrm{MnO}}_{4}^{-}\;\to {\mathrm{Mn}}^{2+}+O_{2}+H_{2}O$$

Finally, we have a solution for the following ions [27]:(11)$${\;H}_{2}O_{2}+{\mathrm{MnO}}_{4}^{-}\;\to {\mathrm{Mn}}^{2+}+O_{2}+H_{2}O$$

### 2.2. GO Reduction

The green synthesis was carried out by preparing graphene oxide solutions exfoliated with the natural extracts of *Larrea Tridentata* and *Capsicum Chinense* at different concentrations.

#### 2.2.1. Description of Plant Phytochemicals

*Larrea Tridentata* is a perennial shrub from the Chihuahua, Sonora and Mohave deserts of North America. The main compounds of *Larrea tridentata* reported in the literature are phenolic lignans, followed by saponins, flavonoids, amino acids and minerals. The most important compound is found in the resin of cells near the upper and lower epidermal layers of the leaves and stems is nordihydroguaiaretic acid (NDGA), one of the better-known antioxidants. Chemically it has been described as beta, gammadimthyl alpha, delta-bis(3,4-dihydroxyphenyl) butane. It has been determined that this acid has antioxidant, anti-inflammatory, cytotoxic, antimicrobial and enzyme inhibitor properties (Figure 3a) [28].

On the other hand, the *Capsicum Chinense* (habanero pepper) is an herbaceous plant with a high source of phytochemicals or bioactive compounds that have shown benefits to human health. The main compounds of *Capsicum Chinense* are phenolic compounds, capsaicinoids, flavonoids, vitamin C, carotenoids and tocopherols. The capsaicinoids present in the chili fruits of the genus *Capsicum Chinense* have various functional properties such as: antioxidant, anti-inflammatory, antimicrobial and antitumor capacity, among others. Additionally, natural capsaicin with a high level of purity is of great economic importance and appreciated in the pharmaceutical, food and chemical industries (Figure 3b) [29].

#### 2.2.2. Preparation of the Extracts

To prepare the gobernadora extract, 30 g of *Larrea’s Tridentata* dry powder was mixed with 100 mL of deionized water at 70 °C for 30 min and a pH value of 5 ± 0.15. The extract obtained was filtered with a 110 mm filter diameter glass microfiber and then centrifuged at 3400 rpm for 15 min. The habanero extract was made in a similar procedure. The aqueous extracts of gobernadora and habanero were prepared at different concentrations of 5%, 10%, and 20% (% *v*/*v*) natural extract—deionized water. The behavior of the extract concentration and it is reducing power in graphene oxide was analyzed.

#### 2.2.3. Reduction of Graphene Oxide

Once the aqueous extracts were prepared, the graphene oxide solutions were exfoliated and mixtures of the aqueous extract solutions with GO were prepared in a ratio of 4:1 (% *v*/*v*). Subsequently, the solutions were heated at 80 °C for 12 h under constant agitation in a reflux system. After 2 h of heating, the color began to gradually change from brown to black, indicating the start of graphene oxide reduction. After the reduction was complete, the color turned dark black and this indicated the reduction of graphene oxide and the formation of reduced graphene oxide (RGO). To complete the green synthesis, the mixtures were centrifuged and washed with deionized water several times and finally dried at 60 °C for further characterization.

### 2.3. Photocatalytic Degradation Studies

For the photocatalytic degradation study, dye selected was methylene blue (MB). UV light (365 nm) irradiation was applied to find out a suitable as well as effective way to eliminate the maximum percentage of dye. For photodegradation, the dye solution was prepared in deionized water at the concentration of 10 mg/L. Further, 20 mg/L of RGO is added to the prepared dye solution. After, these mixtures were subject to low shaking for 20 h under dark conditions for the adsorption and desorption of dye molecules onto the surface of RGO. At the end of the dark exposure period, each mixture was subjected to UV–Vis spectrophotometer analysis and the initial concentration (C_0_) of the solutions was noted before photodegradation. Dye degradation was carried out sequentially by individual dye solutions of MB.

The photodegradation solution was retrieved at regular intervals for the continuous monitoring of the degradation study. The obtained solutions are centrifuged to separate catalysis from the solution and used for UV–Vis Spectrophotometer analysis and noted as Ct (concentration of dye at specific irradiation/exposure time). Dye degradation could be calculated from the absorbance values using the following formula [30,31,32,33];

Percentage of degradation % = (C_0_ − C_t_/C_0_) × 100

### 2.4. Characterizations

The XRD technique allows monitoring the oxidation process of graphite. The diffraction patterns of the GO and RGO were performed in a Miniflex II model, Rigaku brand diffractometer (λ = 1.5418), the measurements of the samples were made in a 2Θ range of 5 to 70°. To determine the reduction of graphene oxide the following techniques were used: UV–Vis, FT-IR and Raman. The interaction of the samples with the electromagnetic radiation was determined by UV–V is spectroscopy with a Perkin Elmer spectrophotometer, scanning wavelengths from 248 to 400 nm, the measurements were made in quartz cells of a 1 cm optical path over aqueous suspensions of graphene The presence of the characteristic bonds as well as the loss of oxygenated groups was determined by means of the FT-IR spectra of the samples with a Perkin Elmer Paragon 1000 spectrometer, the measurements were made in the spectral range between the wave numbers 500 and 4000 cm^−1^. To determine each of the carbon members, Raman spectroscopy using the LabRam VIS-633 equipment was used. The measurements were made in the spectral range between the wave numbers 1000 and 3500 cm^−1^. Finally, the morphology of our material was determined using a transmission electron microscope with a filament of lanthanum hexaboride at a scale of 100 and 200 nm.

## 3. Results

In this work, six samples of a colloidal suspension of graphene oxide (GO) were reduced with different natural aqueous extracts (habanero and gobernadora). Table 1 shows the identification labels for each of the samples.

### 3.1. UV–VIS Spectroscopy

UV–VIS allowed the determination of whether suspensions contained transient products of GO and if it had been partially reduced (RGO) as the natural extract concentrations increased. In this technique only five samples were analyzed: (a) GO, (b) HAB1, (c) HAB2 (d) HAB3, (c) GOB1, (f) GOB2 and (g) GOB3.

During the oxidation of graphite, oxygen adheres to the graphene layers, thus increasing the polarity of the layers which in turn increases its solubility in water. This results in a change in the color of the solution from yellow to brown. depending on the GO concentration, the brown color can be of different intensities. The GO that is formed has an absorption maximum at 225 nm for a well-oxidized material [34]. TheUV–Vis spectra of GO and RGO are given in Figure 4 and Figure 5. This revealed that their optical properties were substantially different. The absorption bands were centered 248 nm from GO to 260 nm from RGO due to the increase in RGO conjugation presumably due to the π and π* transition of the C=C bonds. In samples (d) and (g) (Figure 4 and Figure 5 respectly) a greater shift towards the visible region was observed, assuming that the greater the movement of the absorption spectrum towards the visible region, the better the reduction. This is due to natural reducing agents in plants. Therefore, it can be said that *Larrea Tridentata* extract is a better candidate as a GO-reducing agent compared to *Capsicum Chinense* extract [35,36,37,38].

### 3.2. Infrared Spectroscopy with Fourier Transform

The reduction of the GO leads to the loss of oxygenated functional groups, so it was decided to apply FT-IR spectroscopy to verify if there are differences between the amount and type of functional groups in the samples. The analyzed FT-IR spectra are presented in Figure 6 and Figure 7: (a) GO, (b) HAB1, (c) HAB2, (d) HAB3, (e) GOB1, (f) GOB2 and (g) GOB3.

Of all the FT-IR transmittance spectra of the samples analyzed, peaks attributed to the oxidation of graphite and graphene oxide can be observed (See Figure 6 and Figure 7). Oxidized graphite was characterized by an intense peak that occurs between 3100–3700 cm^−1^ and was attributed to the C-OH stretch bands of the hydroxyl group. The second peak of less intensity was located at approximately 1620 cm^−1^ and was attributed to the skeletal vibration C=C. A third peak was located at approximately 1050 cm^−1^ and was attributed to the C-O alkoxy stretching vibration. Finally, a peak of very low intensity of approximately 1730 cm^−1^ was attributed to the vibration of the carbonyl group C=O [37,38,39]. The next group of peaks contained peaks observed in the graphene oxide spectra. The most intense peak was approximately 995 cm^−1^ and was attributed to the stretching vibration of the epoxy group C–O–C [40]. The next peak was found at approximately 1590 cm^−1^ and was attributed to the skeletal vibration C=C of the graphene planes [41,42]. A third peak with reduced intensity compared to that of graphite oxide at approximately 3400 cm^−1^, was attributed to the stretching vibrations C–OH of a hydroxyl group [43,44,45]. The intensity of this peak was closely related to the oxygen content in the samples tested after reduction with natural extracts. These results, again suggest that as the concentration of natural extracts increased, the GO lost oxygen groups, which led to an increase in sp^2^ carbon bonds, as seen by the increase in the band at 1620 cm^−1^, corresponding to the C=C links. In addition, in all the samples a two-diminution of intensity was observed in the peaks of the functional groups C–O–C and C–OH [17,18,20,37,38,39] (See functional groups in Table 2).

### 3.3. Raman Spectroscopy

Raman spectroscopy is considered a popular technique for characterizing the structural and electronic properties of graphene, including disordered and defective structures, defective density and doping levels. Raman spectroscopy is very sensitive to structure for the characterization of carbon-based materials, especially C=C double bonds that lead to high-Raman intensities. The Raman spectrum of graphene is generally characterized by two main characteristics: the G-peak, which arises from the first-order dispersion of the E_2g_ phonon from sp^2^ carbon atoms (generally observed ~1575 cm^−1^) and the D-peak (~1355 cm^−1^) arising from the mode of respiration of photons of the K symmetry point A_1g_ [37,38,43].

Our results (Figure 8) show that the G-band and the D-band of the GO and the RGO appeared in a range of 1585–1590 cm^−1^ while the D-band appeared in the range of 1342–1350 cm^−1^. Interestingly, the Raman spectrum of GO after reduction with natural extracts had higher intensities than GO, these high intensities could indicate the introduction of sp^3^ defects after functionalization and incomplete recovery of the graphene structure. The variation of the relative intensities of the G-band and the D-band in the GO Raman spectra during the reduction generally reveals the state change of the electronic conjugation. In this work an increase in the number of sp^2^ domains was indicated after GO reduction [17,20,44] (See Table 3).

On the other hand, it is well known that the Raman dispersion of two phonons (2D) of graphene-based materials is a valuable band for differentiating monolayer graphene from double layer or multilayer graphene, since it is highly perceptive to the stacking of graphene layers. Usually, a Lorentzian peak is observed for the 2D band of the monolayer graphene sheets at 2697 cm^−1^, while this peak widens and moves to a larger wave number in the case of multilayer graphene. In this investigation, 2D widened bands were also observed at approximately 2700 cm^−1^ for both the GO and the RGO. This indicates that the GO and RGO samples had a multilayer structure. It is also clarified that after the reduction of GO, the 2D band moved towards a higher value that suggests the stacking of the graphene layers. Since GO has different types of functional groups that can prevent the stacking of graphene layers, but after reduction due to the decrease of such functional groups, some graphene layers are stacked, and form multilayer RGO [18,46]. Figure 5 shows: (a) GO (unreduced), (b) HAB3 RGO-*Capsicum* 20% and (c) GOB3 RGO-*Larrea* 20%.

Table 3 shows the relationship between the intensities of the two peaks (I_D_/I_G_) and will be used to compare the way in which the structural disorder grows in the graphitic network. The locations of both D- and G-bands and the ratio between their intensities are consistent with the characteristic values reported elsewhere. When pristine graphite was oxidized to GrO, the I_D_/I_G_ ratio of the resulting material significantly increased indicating a higher level of structural disorder and a larger number of defects in the graphene layers due to the oxidation process. After sonication to obtain GO, I_D_/I_G_ relationship in the resulting material slightly increased, indicating that the exfoliation process also incorporated further defects into the structure. Logically, the distance between defects (determined as $L_{D}=\sqrt{\frac{C\left(\lambda \right)}{\left(I_{D}/I_{G}\right)}}$ being C(k) = 102 nm^2^) decreased from graphite to both GrO and GO [47,48,49,50,51].

### 3.4. X-ray Diffraction (XRD)

The XRD technique allows for controlling the oxidation process of graphite. Figure 6 shows the powder XRD diffraction patterns of the analyzed samples: the analyzed FT-IR spectra are presented in Figure 9: (a) GO, (b) HAB3 RGO-Capsicum 20%, (c) GOB3 RGO-Larrea 20%.

The data reported in the literature show that graphite has a reflection peak at approximately 2ϴ = 26°, which corresponds to a spacing d of 0.335. Upon oxidizing graphite, the reflection peak (002) changes to the angle 2ϴ = 10° (space d = 0.906). This increase in the d-space is largely due to the intercalation of water molecules and the formation of functional oxygen-containing groups in the graphite layers [38,51,52]. Our GO and RGO samples have a centered peak at 2 ϴ = 26.52°. Corresponding to a spacing d = 0.36 nm that may be due to multilayer graphene [53]. It is also evident that this peak attenuated its intensity considerably in samples (b) and (c), this decrease in intensity is associated with the reduction in GO. The absence of the 2ϴ = 10° peak indicates that the oxygen-containing group of the GO has been removed efficiently. Furthermore, in the three samples, a less intense peak was observed at 2ϴ = 54° which was attributed to the plane (004) [17,18,19,20,47,53].

### 3.5. Transmission Electron Microscopy (TEM)

This characterization technique was used to observe the morphology and size of layers of graphene or graphene oxide, as well as their sizes and imperfections. The samples were prepared by dispersion in alcohol and then sonicated for 10 h. A drop of the suspension was deposited on to a copper grid to analyze it. Figure 10 shows the trans-mission electron microscopy (TEM) of the RGO sample with natural extracts of gobernadora 20%. In Figure 10a shown the graphene sheets at magnification of 100 nm and Figure 10b shown the graphene sheets at magnification of 200 nm. Both the TEM pictures showed good disaggregation of a few layers of RGO samples and the sizes of the sheets in microns.

### 3.6. Methylene Blue (MB) Degradation

Methylene blue (MB), also called methylthionine chloride, is an organic dye with the molecular formulation C_16_H_18_N_3_Cl_1_S_1_ and a molecular weight of 319.85 g/mol. Its maximum absorbance is at 663 nm [54]. The absorbance analysis of the samples after exposure to UV light was performed in the range of 460–750 nm as shown in Figure 11. 

The degradation efficiency (%) was calculated from the absorbance values obtained for the initial dye concentration (C_o_) with the absorbance values obtained at specific time intervals (C_t_) (Figure 12).

The first step of the photocatalysis is the activation of the photocatalyst by the reconstruction of the photocatalyst surface via electromagnetic radiation illumination. The electromagnetic radiations interact with the surface plasmons of the photocatalyst and excite the electrons into the higher states. Upon the excitation of the negatively charged electron (e−), the positively charged hole (h+) is created in its place. These e−/h+ pairs are responsible for the photocatalytic capability of the photocatalyst. These generated e−/h+ can undergo recombination reactions via relaxation processes which lead to the deactivation of the photocatalyst [55,56,57].

From the sample of RGO with gobernadora extract (Figure 11a), it is very clear that within a short time (20 min), absorbance decreased with the increasing time duration. Visually, the loss of color of the MB was also observed, as seen in Figure 11a. On the other hand, in the RGO sample with habanero extract, a loss in absorbance values ~0.45 was observed after leaving the sample in complete darkness under agitation (absorption–desorption), when exposing the sample to UV light we believe that they generate certain recombination reactions, due to the effect of relaxation processes, for which increases and decreases in absorbance were observed as time progressed. (Figure 11b). These dissipation reactions can be prevented if oxidizing agents are incorporated which allow the reduction of the electron-hole pair to promote photocatalytic reaction. [58]. Visually, it was seen that there was no color decrease with increasing time as shown in Figure 12b. Here, a different absorption–desorption process at the RGO-modified surfaces was perhaps obtained, such as can be seen in Figure 11b, demonstrating an increase and decrease in the percentage of MB.

The efficiency of photocatalytic degradation of MB with a photocatalyst was calculated from absorbance values and is presented in Figure 13. It was observed that the RGO sample with gobernadora extract showed 89.8% (in 120 min) degradation, and the RGO sample with habanero extract showed 60.3% (120 min) degradation.

The previously studied reduced graphene was compared with other studies on the removal of MB dye from similar light sources (Table 4). Numerous researchers have evaluated GO and RGO using photocatalytic materials known as composites for photocatalyst enhancement. For example, Patidar et. Alabama, 2018, carried out RGO with CdSe and obtained only 56–63% MB degradation efficiency after 240 min of UV illumination. Many other GO and RGO nanocomposite materials showed more than 90% degradation efficiency with 120–180 min of exposure time. In the present study, RGO–GOB3 showed considerable photocatalytic activity within 20 to 60 min of exposure. We suggest that the full degradation performance can be obtained if the exposure time is longer.

## 4. Discussion

Based on these results, we obtained that the concentrations of the extract directly affect the reduction of our material, so we conclude that the higher the concentration of the natural extract (20% *v*/*v*), the better the reduction of GO. Using Raman spectroscopy, we noticed that both GO and RGO obtained had a multilayer structure, possibly due to the lack of exfoliation time in the ultrasound period before and after the graphene oxide reduction. Prior to characterization by TEM microscopy, RGO was exfoliated for 10 h and monolayers were obtained. The TEM images reaffirmed the multilayer structure in RGO samples prepared by our eco-friendly route of synthesis.

Both extracts functioned as GO-reducing agents. However, due to the results obtained from the UV–Vis, FT-IR and XRD characterizations, we concluded that the *Larrea Tridentata* extract was the best reducing agent due to its high content of chlorogenic acid, caffeic acid, p-coumaric acid, and ferulic acid [67], favoring the reduction and structural arrangement of the hydroxyl groups of GO compared to those of habanero extract (phenolic compounds, capsaicinoids, and ascorbic acid).

On the other hand, according to characterizations by UV–VIS spectrophotometer analysis, it is assumed that the GO reduced by *Larrea Tridentata* extract is an optimal photocatalyst for MB dye degradation with a % degradation efficiency of 90%.

## 5. Conclusions

Reduced graphene oxide (RGO) was obtained through an eco-friendly synthesis applying a green chemistry methodology and using natural extracts of *Larrea Tridentata* (gobernadora) and *Capsicum Chinense* (habanero). We believe that the reduction is largely due to natural-reducing agents such as ascorbic acid, carotenes, polyphenols and capsicum contained in plants. In addition, we can also say that the *Larrea Tridentata* extract acts as the best GO reducer, according to the results obtained from the FT-IR, UV–Vis and XRD characterization techniques. On the other hand, it was possible to verify that the RGO reduced with natural extracts can act as a photocatalyst for the degradation of MB dye [68,69,70,71].

The use of eco-friendly reducing agents to synthesize graphene garners great interest thanks to the great potential of applications that they can have. It is considered as one of the most versatile methods. These kinds of reagents are shown to be an alternative to inorganic compounds, such as hydrazine or other non-biocompatible or toxics reducing agents, as well as with better catalytic properties compared to other organic reducers, such as glucose. They are also safe to handle and the reaction co-products are biocompatible.

## Figures and Tables

**Figure 1 nanomaterials-12-03882-f001:**
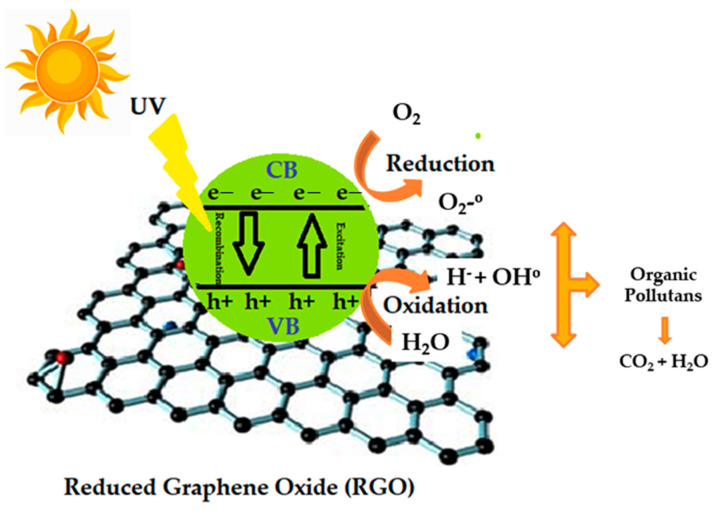
General scheme of photocatalysis.

**Figure 2 nanomaterials-12-03882-f002:**
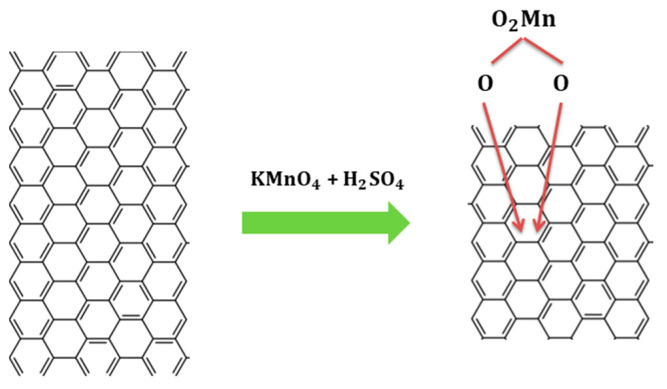
Scheme of the permanganate behavior in the presence of sulfuric acid.

**Figure 3 nanomaterials-12-03882-f003:**
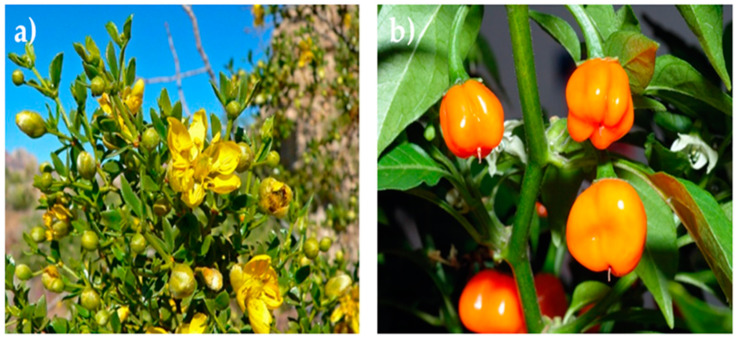
Plants of (**a**) Larrea Tridentata (gobernadora) and (**b**) Capsicum Chinense (habanero).

**Figure 4 nanomaterials-12-03882-f004:**
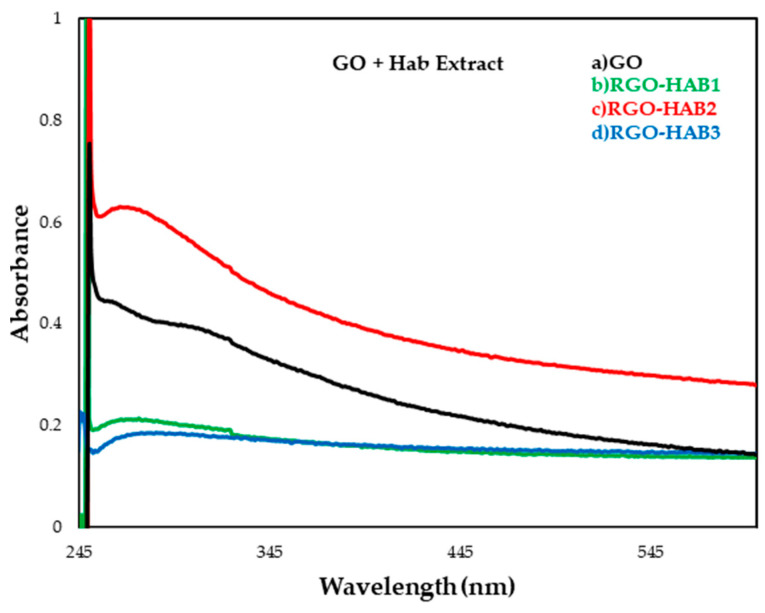
UV–VIS absorption spectrum of graphene oxide (GO) and reduced graphene oxide (RGO) samples with natural extracts: (a) GO (unreduced), (b) RGO-HAB1 5%, (c) RGO-HAB2 10% (d) RGO-HAB3 20%.

**Figure 5 nanomaterials-12-03882-f005:**
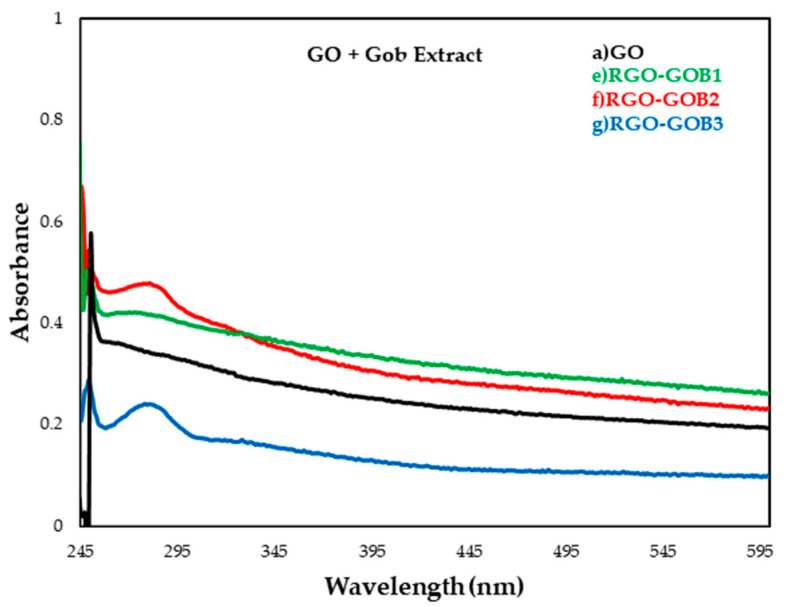
UV–VIS absorption spectrum of graphene oxide (GO) and reduced graphene oxide (RGO) samples with natural extracts: (a) GO (unreduced), (e) RGO-GOB1 5% (f) RGO-GOB2 10% and (g) RGO- GOB3 20%.

**Figure 6 nanomaterials-12-03882-f006:**
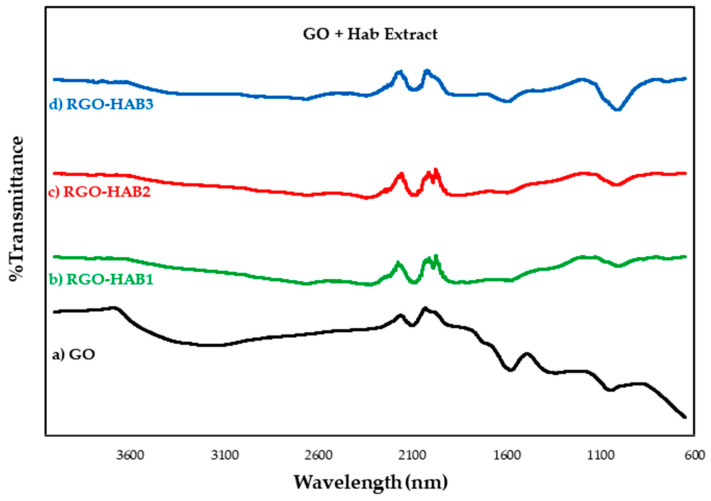
FT-IR transmittance spectrum of graphene oxide (GO) and reduced graphene oxide (RGO) samples with natural extracts. The analyzed samples are (a) GO (unreduced), (b) RGO-HAB1 5%, (c) RGO-HAB2 10% (d) RGO-HAB3 20%.

**Figure 7 nanomaterials-12-03882-f007:**
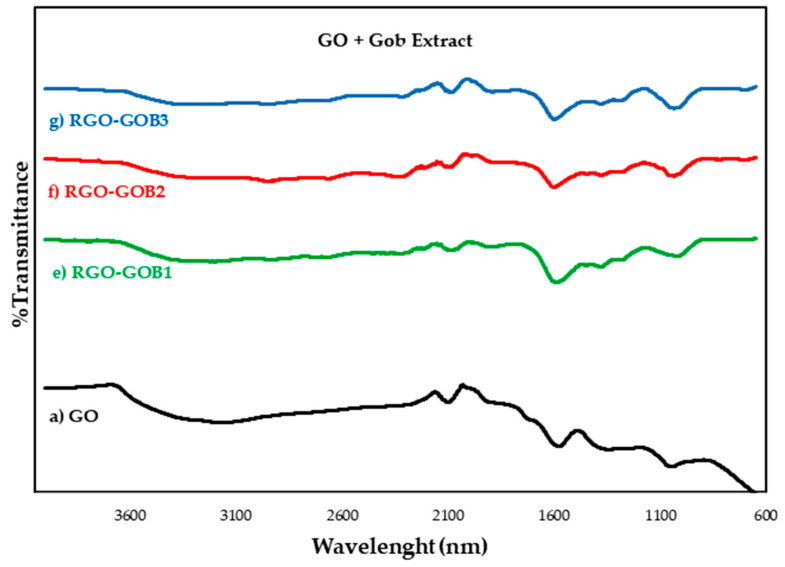
FT-IR transmittance spectrum of graphene oxide (GO) and reduced graphene oxide (RGO) samples with natural extracts. The analyzed samples are (a) GO (unreduced), (e) RGO-GOB1 5% (f) RGO-GOB2 10%, (g) RGO-GOB3 20%.

**Figure 8 nanomaterials-12-03882-f008:**
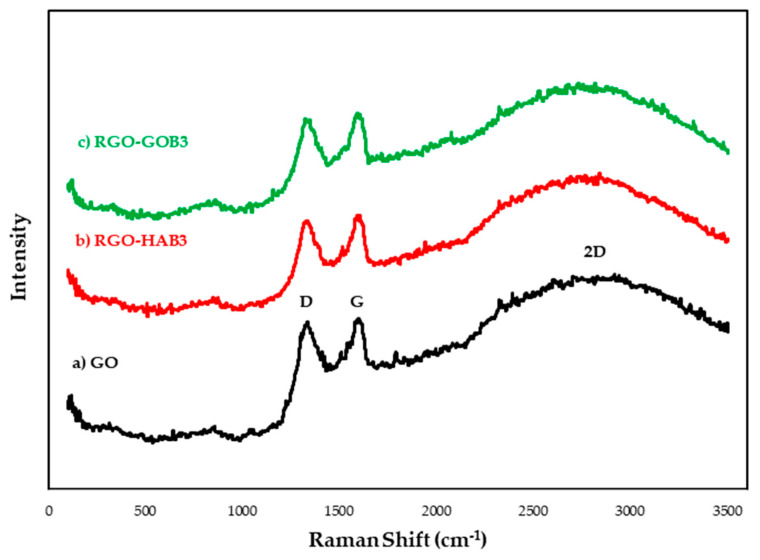
Raman spectrum of GO and RGO with natural extracts. The analyzed samples are: (a) GO (unreduced), (b) RGO-HAB3 20% and (c) RGO-GOB3 20%.

**Figure 9 nanomaterials-12-03882-f009:**
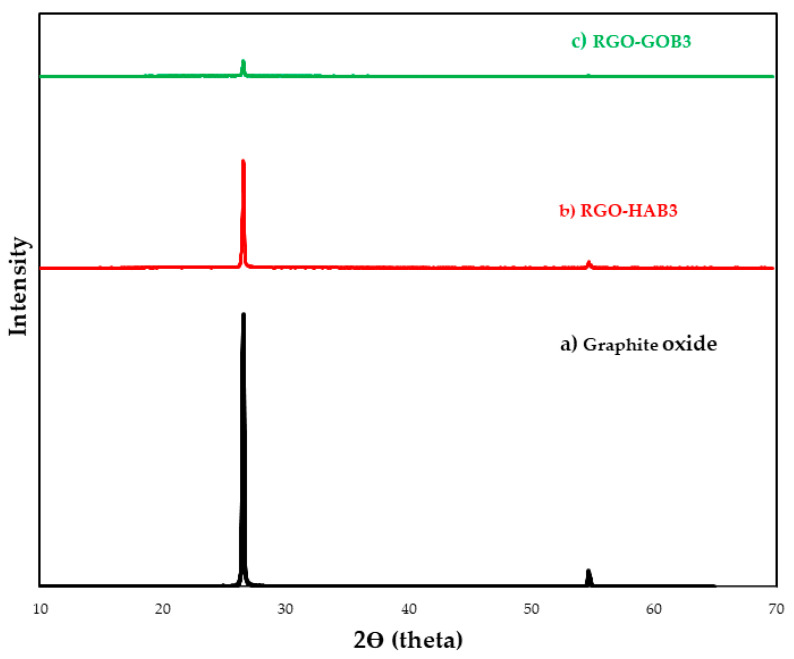
XRD diffraction patterns of graphene oxide (GO) and reduced graphene oxide (GRO) with natural extracts. (a) GO (unreduced), (b) HAB3 RGO-Capsicum 20%, and (c) GOB3 RGO-Larrea 20%.

**Figure 10 nanomaterials-12-03882-f010:**
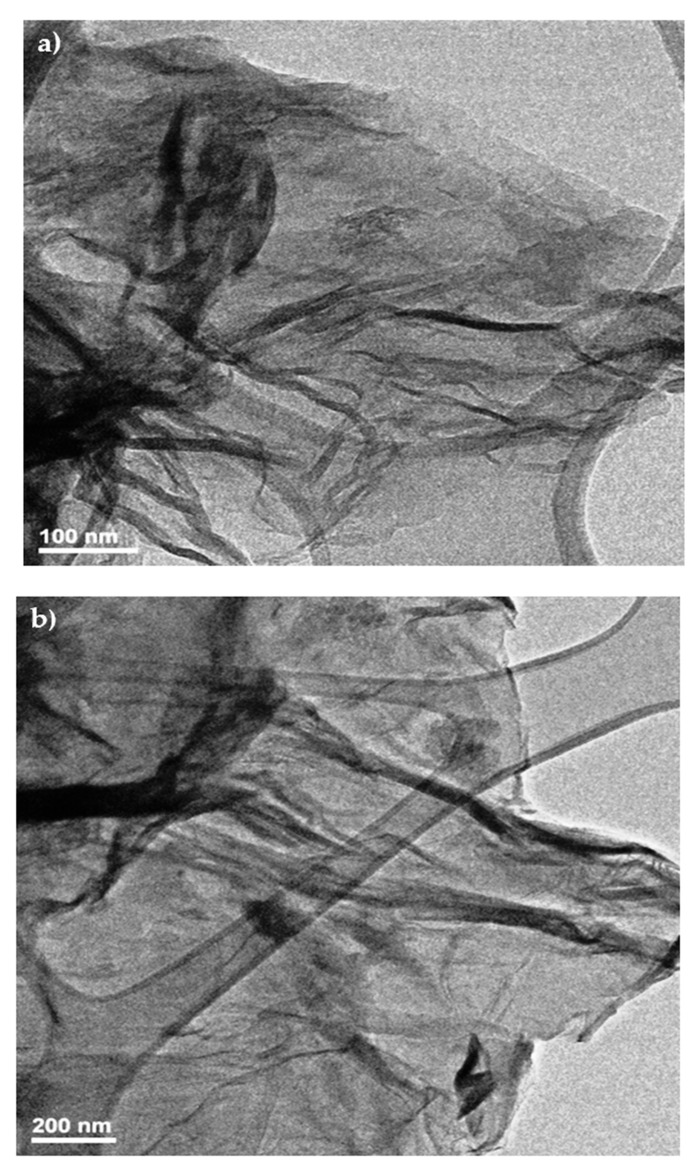
High-resolution TEM microscopy of graphene oxide reduced with natural extracts at (**a**) magnification of 100 nm and (**b**) magnification of 200 nm.

**Figure 11 nanomaterials-12-03882-f011:**
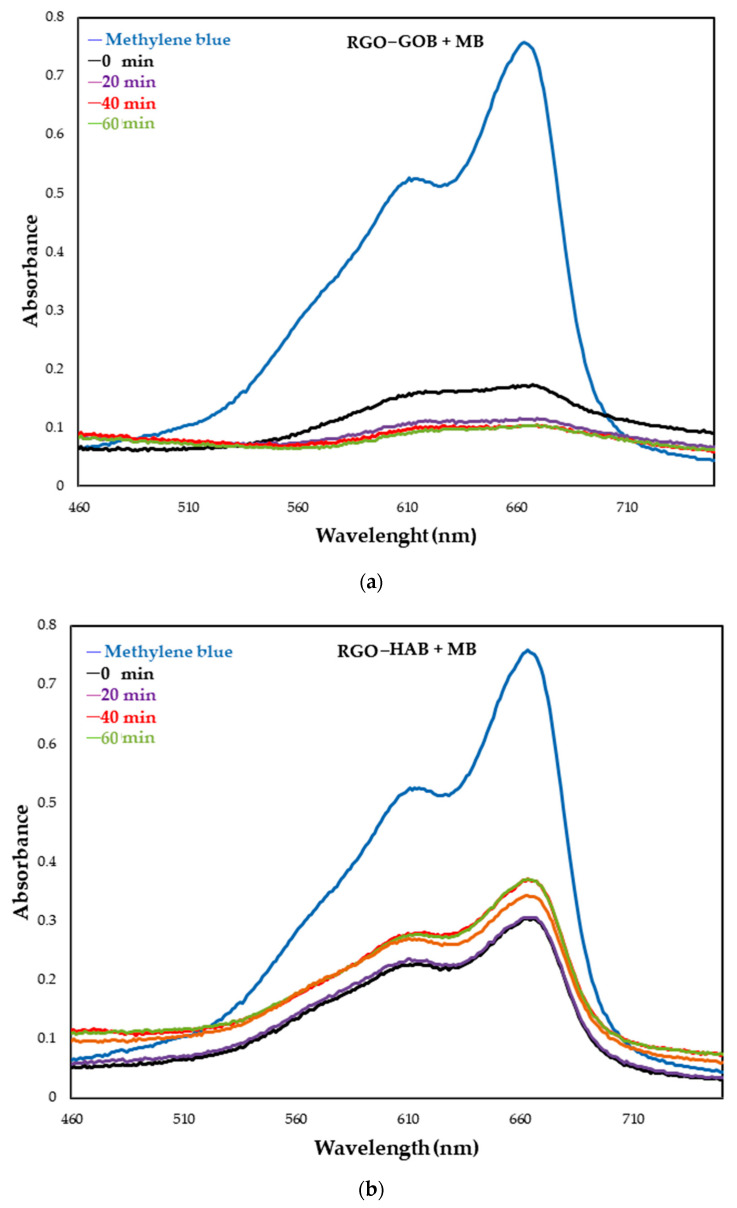
UV–Vis absorption spectra of MB dye solution at different time intervals in the presence of (**a**) ROG–GOB3 and (**b**) RGO–HAB3 under UV light exposure.

**Figure 12 nanomaterials-12-03882-f012:**
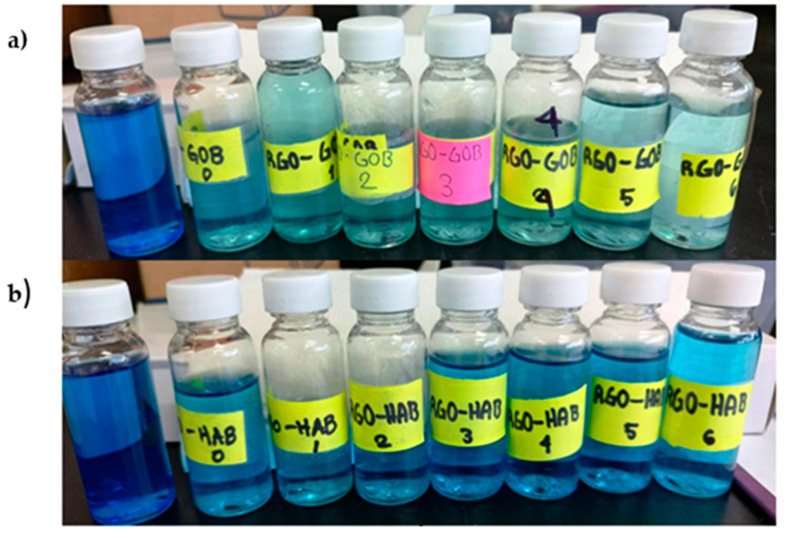
Degradation of dye (**a**) RGO with gobernadora extract and (**b**) RGO with habanero extract.

**Figure 13 nanomaterials-12-03882-f013:**
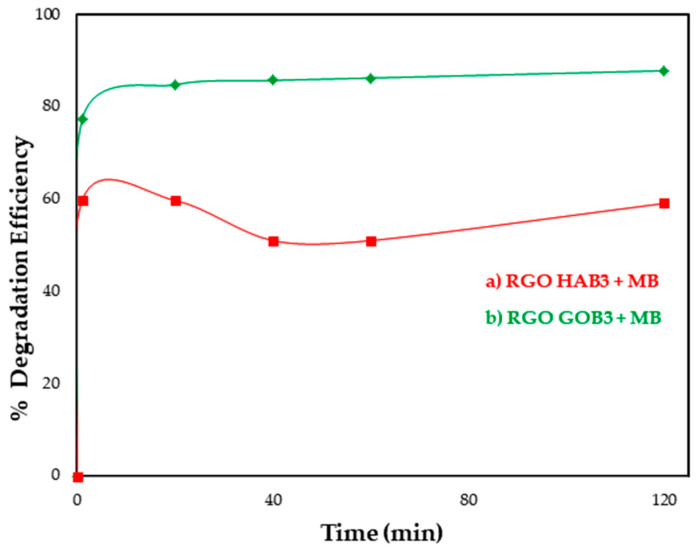
Degradation efficiency (%) of (a) RGO with habanero extract and (b) RGO with gobernadora extract.

**Table 1 nanomaterials-12-03882-t001:** Identification labels for each sample of reduced graphene oxide.

Concentrations of the Natural Extract	Extract 1 (*Capsicum Chinense*)	Extract 2 (*Larrea Tridentata*)
0%	GO	GO
5%	HAB1	GOB1
10%	HAB2	GOB2
20%	HAB3	GOB3

**Table 2 nanomaterials-12-03882-t002:** Characteristic FTIR absorption bands of oxidized graphite and reduced graphene oxide (RGO).

Wavelength (cm^−1^)	Functional Group		Oxidized Graphite	RGO
3100–3700	(Hydroxil group)	C–OH	Yes	Yes
1730	(Ester, aldehyde and carboxil acid groups)	C=O	Yes	No
1620	(Alkene group)	C=C	Yes	No
1590	(Alkene group)	C=C	No	Yes
1050	(Alcoxy group)	C–O	Yes	No
995	(Epoxy group)	C–O–C	No	Yes

**Table 3 nanomaterials-12-03882-t003:** Raman characteristic parameters of graphite, graphene oxide (GO) and reduced graphene oxide (RGO) samples.

Raman	I_D_/I_G_	$L_{D}=\sqrt{\frac{C\left(\lambda \right)}{\left(I_{D}/I_{G}\right)}}$	References
Graphite	0.19	23.169	[47,48,49]
GO	0.99	10.150	[47,48,49,50]
RGO-HAB3	0.987	10.166	This work
RGO-GOB3	0.983	10.186	This work

**Table 4 nanomaterials-12-03882-t004:** Comparison of photocatalysts with their photocatalytic performance against MB dye removal under similar light sources.

Catalyst	% of Degradation	Time (min)	Reference
CdSe–RGO	56%	210	[59]
RGO–HAB3	60.3%	120	This work
GO–Clay nanocomposite	87%	30	[60]
RGO–NiFe	88.7%	300	[61]
RGO–GOB3	89.8%	120	This work
RGO–ZnO	91%	60	[62]
BaTiO_3_–GO	95%	180	[63]
Graphene–Ag nanocomposite	96%	120	[64]
MnFe_2_O_4_–RGO	97%	60	[65]
RGO–Fe_3_O_4_/TiO_2_	99%	55	[66]

## Data Availability

Not applicable.

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
