# Peer review of "Eco-Friendly Reduction of Graphene Oxide by Aqueous Extracts for Photocatalysis Applications"

_nanomaterials, 2022, doi:10.3390/nano12213882_

Round 1
Reviewer 1 Report
On my opinion the paper is not well written, for these reason it can not be published in the present form. I recommend major revision of the manuscript, based on the following comments;
In this title, photocatalysis applications is in the focus, however, photocatalysis applications does not emerge in the manuscript. On the other hand, eco-friendly is mentioned in the title. the eco-friendly of synthesis methods need to be supplemented.
The novelty of the paper needs to be clearly stated at the end of section 1
As the aim of the study is to eco-friendly reduction of graphene oxide, the advantages of graphene oxide as materials should be stressed in the introduction part with more convincing references.
If possible, comparisons between the photocatalysis applications performance of this study and other studies on photocatalysis should be presented.
Author Response
In response to the reviews, the article is modified according to the comments received, and an application is added for the photocatalytic evaluation of the proposed material.
The green synthesis processes are described in detail and the advantages compared to other more toxic methods are referred to and a comparative table between catalysts applied to the degradation of methylene blue is added.
1. On my opinion the paper is not well written, for these reason it can not be published in the present form. I recommend major revision of the manuscript, based on the following comments;
In this title, photocatalysis applications is in the focus, however, photocatalysis applications does not emerge in the manuscript. On the other hand, eco-friendly is mentioned in the title. the eco-friendly of synthesis methods need to be supplemented.
The manuscript was modified giving priority to the eco-friendly method and to the application in photocatalysis for the degradation of the methylene blue dye.
2. The novelty of the paper needs to be clearly stated at the end of section 1
The novelty of the green method has been developed at the end of section 1.
3. As the aim of the study is to eco-friendly reduction of graphene oxide, the advantages of graphene oxide as materials should be stressed in the introduction part with more convincing references.
New references have been added to support this work.
4. If possible, comparisons between the photocatalysis applications performance of this study and other studies on photocatalysis should be presented.
A comparative table of photocatalysts with RGO and other materials is added and their degradation efficiencies are reviewed.

Reviewer 2 Report
The subject of the paper is interesting and the results of the study are important for improving the efficiency of graphene oxide reduction by using green agents. Plant extracts (Larrea Tridentata and Capsicum Chinense) turned out to be good alternative to organic or inorganic reducing agents.
The authors should explain why the use of plant extracts derived from Larrea Tridentata and Capsicum Chinense is more beneficial and green than the use of, for instance, glucose that can be easily produced from cellulose by fermentation.
The quality of reproduced figures (Figs. 1, 2, 4, 6) is very poor and new graphs should be given.
Figure 2 does not present any mechanisms of oxidation as it is claimed in the title, but rather the locations of the oxidation sites in the graphene structure.
The X-axis of Fig. 3 is obviously related to wavelengths, not wavenumbers. So the authors should correct either the axis by placing the wavenumbers or the caption of the axis (Wavelength, nm). Also, it is always expected that the values for the Y axis (Absorbance) increase upward, while the spectra presented looks like mirrors (absorbance increases downward). It looks like the same happened with Fig. 4 (Transmittance mode).
Figure 6: XRD diffraction patterns, not spectra.
Author Response
All corrections have been added to the article.
1. The subject of the paper is interesting and the results of the study are important for improving the efficiency of graphene oxide reduction by using green agents. Plant extracts (Larrea Tridentata and Capsicum Chinense) turned out to be good alternative to organic or inorganic reducing agents.
The natural extracts of Larrea Tridentata and Capsicum Chinense are an alternative for the reduction of GO with respect to conventional inorganic chemical reducers such as hydrazine.
2. The authors should explain why the use of plant extracts derived from Larrea Tridentata and Capsicum Chinense is more beneficial and green than the use of, for instance, glucose that can be easily produced from cellulose by fermentation.
These extracts are beneficial compared to other natural agents such as glucose for the following reasons:
- a) These extracts are made simply, quickly and cheaply.
- b) The properties of the plants allow to obtain stable structures of the RGO and to compare the reduction of the GO with other inorganic reducers (such as hydrazine), but these extracts are friendly to the environment.
- c) The reduction of GO with glucose is considered a mild reduction, so it is not possible to efficiently remove the C-OH groups from GO. These extracts show considerable results in the reduction of the GO hydroxyl group after reduction.
3. The quality of reproduced figures (Figs. 1, 2, 4, 6) is very poor and new graphs should be given.
All graphs and figures have been modified.
4. Figure 2 does not present any mechanisms of oxidation as it is claimed in the title, but rather the locations of the oxidation sites in the graphene structure.
Figure 2 has been removed.
5. The X-axis of Fig. 3 is obviously related to wavelengths, not wavenumbers. So the authors should correct either the axis by placing the wavenumbers or the caption of the axis (Wavelength, nm). Also, it is always expected that the values for the Y axis (Absorbance) increase upward, while the spectra presented looks like mirrors (absorbance increases downward). It looks like the same happened with Fig. 4 (Transmittance mode).
The axes in the graphs have been modified
6. Figure 6: XRD diffraction patterns, not spectra.
The title of the graph was changed to diffraction patterns.

Round 2
Reviewer 1 Report
My only remark is the low quality of figures 3 -10. They should be redrawn.
Author Response
Hi, I send the final version of the manuscript. I modified the format of the figures. Thank for the revision.

Reviewer 2 Report
The authors improved the manuscript by addressing all the comments
Author Response
Hi, I send the final version of the manuscript. Thank for the revision.
